# Non-Uniform Adversarially Robust Pruning

**Qi Zhao**[1] **Tim Königl**[1] **Christian Wressnegger**[1]

[1]KASTEL Security Research Labs, Karlsruhe Institute of Technology (KIT), Germany

**Abstract**    Neural networks often are highly redundant and can thus be effectively compressed to a fraction of their initial size using model pruning techniques without harming the overall prediction accuracy. Additionally, pruned networks need to maintain robustness against attacks such as adversarial examples. Recent research on combining all these objectives has shown significant advances using uniform compression strategies, that is, parameters are compressed equally according to a preset compression ratio. In this paper, we show that employing *non-uniform compression strategies* allows to improve clean data accuracy as well as adversarial robustness under high overall compression—in particular using channel pruning. We leverage reinforcement learning for finding an optimal trade-off and demonstrate that the resulting compression strategy can be used as a plug-in replacement for uniform compression ratios of existing state-of-the-art approaches. Our code is available at https://intellisec.de/research/heracles

## 1 Introduction

Deploying deep neural networks on resource-constrained hardware is often hindered by the sheer size of the network. Neural network pruning effectively removes redundancy at different structural granularity to reduce a model's size. In safety-critical environments, these networks additionally need to be robust against attacks, such as adversarial examples (Szegedy et al., 2014). With adversarial training (Madry et al., 2018; Shafahi et al., 2019; Wong et al., 2020) it is possible to significantly improve robustness by introducing adversarial examples into the training process. However, recent research (Zhang et al., 2019) suggests that large networks have higher adversarial robustness. Consequently, it is inherently difficult to strike a balance between the compactness and robustness against attacks when pruning neural networks.

The typical network pruning procedure consists of three stages (Liu et al., 2019): First, an over-parameterized model is trained. Second, this pre-trained model is pruned based on a specific criterion and strategy. Finally, the pruned network is fine-tuned to recover the potentially lost performance. The most critical step in the procedure is the second one that defines the pruning objective and any additional objectives next to network compression itself. Han et al. (2015) propose to prune network connections following the order of weight magnitude (OWM), which later on has also been shown effective for robustness-aware pruning by Sehwag et al. (2019). Ye et al. (2019) and Gui et al. (2019) inherit this criterion and define network pruning as an optimization problem that can be solved by the alternating direction method of multipliers (ADMM), initially proposed by Boyd et al. (2011). Similarly, Sehwag et al. (2020) formulate the pruning criterion as an importance score-based optimization problem that, however, anchors adversarial robustness deeply in the pruning process itself. While both OWM (Ye et al., 2019) and optimization-based criteria (Sehwag et al., 2020) yield good results for robust-aware pruning, they require the specification of the compression ratio as an hyper-parameter that is then used uniformly across all layers. Madaan et al. (2020) propose ANP-VS to combine adversarial training with pruning and thus they merge the previously mentioned steps one and two. As such, the method pursues a different goal for which compression does not need to be adjustable. However, ANP-VS learns an implicit non-uniform compression that yields promising results.

In this paper, we follow this intuition and investigate the possibility of improving both compression and adversarial robustness of existing state-of-the-art approaches using *non-uniform compression strategies*. The necessity of non-uniform compression is most evident for channels for which Table 1 provides a first glimpse of the improvements made by our method HERACLES. We prune a network's layers based on the order of weight magnitude (OWM), but determine the compression rate per layer. Inspired by He et al. (2018), we leverage deep reinforcement learning (Deep-RL) to automatically find this global pruning strategy to yield an optimal trade-off between accuracy and adversarial robustness of the pruned network. The determined compression strategy is then used with approaches for pruning a pre-trained model which allows for increasing accuracy on benign as well as adversarial inputs.

| Method | Channel Compr. | Acc. on Benign Data | | Acc. on Attack Data | |
| --- | --- | --- | --- | --- | --- |
| | | Uniform / Non-Uniform [ % ] | | Uniform / Non-Uniform [ % ] | |
| HYDRA | 0.50 | 69.92 / 76.82 | +6.90 | 39.82 / 47.06 | +7.24 |
| | 0.10 | 10.00 / 59.83 | +49.83 | 10.00 / 38.70 | +28.70 |
| R-ADMM | 0.50 | 72.65 / 77.59 | +4.94 | 43.60 / 46.16 | +2.56 |
| | 0.10 | 56.24 / 67.04 | +10.80 | 32.65 / 41.38 | +8.73 |

Table 1: Uniform vs. non-uniform pruning of channels for VGG16 on CIFAR-10.

**Contributions and impact**. We show that a *non-uniform*, global compression strategy is beneficial for effective network pruning when considering adversarial robustness. The compression strategy learned by HERACLES can be applied to state-of-the-art pruning techniques as a plug-in replacement for manually specified compression rates to improve original (benign) and adversarial accuracy whilst yielding the same overall compression. In extensive experiments with the CIFAR-10, SVHN, and ImageNet datasets, we show to surpass the performance of Robust-ADMM (Ye et al., 2019) and HYDRA (Sehwag et al., 2020) in pruning channels that originally use uniform compression strategies. As shown in Table 1, for channel pruning on VGG16, we yield up to 10.80 percentage points higher begin accuracy and 8.73 percentage points higher accuracy under adversarial inputs using Robust-ADMM. For HYDRA, we even successfully escape from a completely damaged model and achieve a remarkable performance improvement. In summary, we are able to significantly improve channel pruning over related work and maintain at least on-par performance in weight pruning. In practice, the additionally gained model robustness helps increase the security and safety of applications on hardware-constraint platforms, for instance, in autonomous driving or edge AI. Moreover, channel pruning is particularly suitable for hardware deployment as it straightforwardly reduces the dimensionality of the necessary computations and thus also speeds up inference.

## 2 Background

We begin by briefly recapping concepts that are central to our approach, such as basic background on network pruning, adversarial training, and reinforcement learning.

### 2.1 Network Pruning

Network pruning enables to compress over-parameterized neural networks by removing structural redundancy (Han et al., 2015, 2016). For this, usually a binary mask $M$ with elements in $\{0, 1\}$ is introduced to cancel out redundant network connections at weight level or channel level. We represent this masking operation by the Hadamard product $\odot$ that transforms the model (its parameters) at the $l$ th layer of the network, $\theta^{(l)}$, to a sparse (pruned) representation $\widetilde{\theta}^{(l)}$:

$$\widetilde{\theta}^{(l)} = M^{(l)} \odot \theta^{(l)}.$$

Note that determining the importance of connections, and thus populating the binary mask $M$, depends on the criterion used in the pruning stage. The order of weight magnitude (OWM) has

been shown to outperform other criteria such as Variational Dropout (Molchanov et al., 2017), Soft Weight-Sharing (Karen Ullrich, 2017), or Filter Standard Deviation (Sun et al., 2019). Thus, it is seen as the gold standard in network pruning (Liu et al., 2019). Consequently, for Heracles, we pick up the OWM criterion for pruning as well but learn a global strategy. Similar to Sehwag et al. (2020), we use scored masks to binarize the pruning mask and initially assign scores to each element of the pruning masks $M$ based on scaled-absolute-initialization:

$$\psi^{(l)} = \frac{|\theta^{(l)}|}{\max |\theta^{(l)}|} \, ,$$

where $|\theta^{(l)}|$ takes the absolute values of model's parameters of layer $l$. Note that for channel pruning, score masks are commonly initialized by the sum of absolute weights along each channel to comply with the OWM criterion.

## 2.2 Adversarial Training

To date, adversarial training (Madry et al., 2018) in its different manifestations (Zhang et al., 2019; Wong et al., 2020) is the most efficient defense against adversarial examples (Szegedy et al., 2014). It generates attacks and incorporates them in the training process, solving a min-max optimization problem, which is formally expressed as:

$$\min_{\theta} \mathbb{E}_{(x,y)\sim \mathcal{D}_t} \left[ \max_{\delta} \mathcal{L}_{adv}(\theta, x + \delta, y) \right].$$

Input pairs of a data sample $x \in \mathbb{R}^d$ and its label $y \in [k]$ are drawn from the training data distribution $\mathcal{D}_t$, where $k$ represents the number of classes. As the normal training procedure, the *outer minimization* reduces the loss function $\mathcal{L}_{adv}$, for instance, the cross-entropy loss. The *inner maximization* is formulated to increase the maximally allowed (adversarial) perturbation $\delta$ for each input data sample $x$, and is solved by projected gradient descent (PGD) (Madry et al., 2018). Building on top of this concept, several approaches have been proposed that improve upon the performance of PGD-based adversarial training (Zhang et al., 2019; Shafahi et al., 2019; Wong et al., 2020).

However, Guo et al. (2018) and Ye et al. (2019) show that increasing adversarial robustness is accompanied with stronger parameter distribution, which commonly hinders network pruning. By striving for a globally optimal compression strategy with varying compression ratios per layer, we show that adversarial robustness and large compression rates are not mutually exclusive.

## 2.3 Reinforcement Learning

For reinforcement learning (RL), an agent strives for an action strategy to maximize the reward $\mathcal{R}$ over multiple episodes $i$ that provides feedback about the effectivity of certain actions in a specific environment (Sutton and Barto, 2018):

$$\max_{\pi} \mathbb{E} \left[ \sum_{i=0}^{\infty} \gamma^i \, \mathcal{R}_i \mid s_0 = s \right].$$

Here, $s$ refers to the agent's state and $\gamma$ represents the discount rate in each episode $i$. Policy $\pi$ aims to maximize the cumulative reward by optimizing the mapping from states to actions taken by the RL agent. To tailor this process to a particular application, such as network pruning, we have to define a *state space* representing the environment as well as an *action space* that specifies allowed actions. The agent then outputs a so-called action space vector to influence its "location" in the environment. In our case, this environment is the model $\theta$ we operate on.

For instance, Huang et al. (2018) deploy a RL agent for a filter pruning, where the state space is composed of the number of input feature maps and the shape in each filter. The agent returns a discrete action vector that scores the importance of each filter.

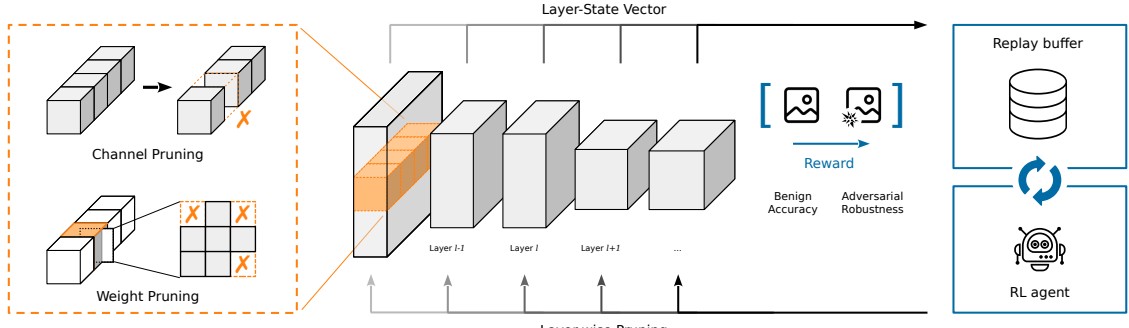

Figure 1: Schematic depiction of Heracles, with an intuition of channel and weight pruning (left) and the global composition of the layer's state (center) as used for reinforcement learning (right).

## 3 Non-Uniform Adversarially Robust Pruning

Heracles searches for the *globally optimal pruning strategy* that increases compression of an adversarially trained network with minimal degradation on both benign accuracy and adversarial robustness. A schematic depiction is provided in Fig. 1. In contrast to focusing on direct connections in a network as implemented in related work (Sehwag et al., 2020), we consider the relations of all layers to each other (pre and post relations) and observe that these far-reaching contexts effect the robustness after pruning and fine-tuning. Finding an optimal compression strategy $[a^{(1)}, \ldots, a^{(L)}]$ under these constraints for all $L$ layers is challenging and is best solved automatically.

**Model compression**. We consider $\Theta^{(l)}$ as the $l$ th layer's total number of parameters and define the compression rate $a^{(l)}$ as the ratio of preserved parameters $\Theta^{(l)}_{saved}$ to all parameters of layer $l$,

$$a^{(l)} = \frac{\Theta^{(l)}_{saved}}{\Theta^{(l)}}.$$

The compression rate for the entire network, $a$, is computed analogously. In line with He et al. (2018), we make sure that the network is not compressed below a specified global compression rate, $a_{min}$. For this, the layer-specific compression rate $a^{(l)}$ is constrained to ensure that the overall compression is *lower than* the sum of i) already pruned parameters of all layer up to $l - 1$, $\widetilde{v}$, ii) parameters that are about to be pruned in layer $l$, $\widetilde{\theta}^{(l)}$, and iii) potentially removed parameters by the most aggressive compression rate $a_{min}$ in layers from $l + 1$ onward. This mechanism allows to precisely control the network's size. For Heracles, we additionally adapt the action range to fit different network compression rates and allow for weight as well as channel pruning. In Appendix A.1, we provide further details on the process.

### Learning Globally Optimal Compression

Based on the above definition of network compression, we resume to define the details of learning a globally optimal compression strategy using reinforcement learning (Algorithm 1). At each iteration of the searching process, we prune the model as outlined in Section 2.1 to determine the accuracy and robustness of the current state. In the following, we detail the definition of the state and action space, specify the reward function used, and elaborate on the exploration phase.

**State space**. For reinforcement learning, we define the RL state $s^{(l)}$ for layer $l$ based on the following eleven features:

$$\left(l, c_{in}, c_{out}, h, w, k, stride, \Theta^{(l)}, \widetilde{v}, v, a_{prev}\right).$$

All features but the compression rate of the previous layer, $a_{prev}$, are dependent on layer $l$: For instance, the $l$ th layer and its output have shape $k \times k \times c_{in} \times c_{out}$ and $h \times w \times c_{out}$, respectively. *stride* refers to the striding offset used for convolutional layers, which may vary depending on input size of subsequent layers. Additionally, we use $\Theta^{(l)}$ to denote the number of parameters of a specific layer, and specify the number of compressed parameters, $\widetilde{v}$, that are produced by pruning so far in preceding layers, as well as parameters remaining in latter layers, $v$. Moreover, we normalize all states to avoid overfitting.

---

**Algorithm 1** HERACLES' non-uniform strategy search

---

**Input**: Pretrained Model $\theta$, The number layers $L$, RL-Agent $RLA$, Target rate $a_{target}$, Rate range $[a_{min}, a_{max}]$, Warm-up episodes $N_{wup}$, Search episodes $N_{srch}$, Valid-set $\mathcal{D}_{val}$
**Output**: Global optimal non-uniform strategy $[a^{(1)}, ..., a^{(L)}]$

---

1: Mask scores initialization: $\psi = \frac{|\theta|}{\max(|\theta|)}$
2: **for** Episode = 1 ... $N_{srch}$ **do**
3:     **for** $l$ = 1 ... $L$ **do**
4:         **if** Episode $\leq N_{wup}$ **then**
5:             $a^{(l)} = random\_uniform(0, 1)$                 # Use random compression rate
6:         **else**
7:             Train RL-Agent with sampled data
8:             $a^{(l)} = \mathcal{N}_{trunc}\left(RLA(s^{(l)}), \sigma^2, 0, 1\right)$          # Predict compression rate
9:         **end if**
10:       $a^{(l)} = a_{min} + a^{(l)} \cdot (a_{max} - a_{min})$               # Re-scale rate
11:       $a^{(l)}_{allow} = \text{Max-Allow-Action}(a^{(l)}, a_{target})$     # Compute maximal allowed rate
12:       $a^{(l)} = \min(a^{(l)}, a^{(l)}_{allow})$             # Action control by $a_{allow}$
13:       $M^{(l)} = \mathbf{1}\left(\psi^{(l)} \geq \psi^{(l)}_K\right)$      # Binary mask transformation with $a^{(l)}$
14:       $\widetilde{\theta}^{(l)} = M^{(l)} \odot \theta^{(l)}$                 # Layer pruning
15:     **end for**
16:     Robustness evaluation on $\widetilde{\theta}$ with $\mathcal{D}_{val}$
17: **end for**

---

**Action space**. The action space of the RL agent here is (roughly speaking) the range of valid compression ratios. In contrast to prior work (Huang et al., 2018), we do not directly produce a discrete binary mask for all layers, but use the Deep Deterministic Policy Gradient (DDPG) algorithm (Lillicrap et al., 2016) to predict a continuous compression rate along each layer. This allows us to approach finer granularity and prune layers that have different shapes. Consequently, the action space used for HERACLES is in the range of $(0, 1]$.

To facilitate more stable reinforcement learning, we use a replay buffer that is initialized in the RL agent's warm-up stage using a random uniform distribution to generate $a^{(l)}$ (line 5). In the exploration-exploitation stage of the RL process, we then use a truncated normal distribution to add noise to the action predicted by the RL agent ($RLA$) with $\sigma = 0.5$ which exponentially decays with each episode (line 8):

$$\mathcal{N}_{trunc}\left(RLA(s^{(l)}), \sigma^2, 0, 1\right)$$

Further details on the action range and the action control algorithm are specified in Algorithm 2 of Appendix A.1, where we introduce the used thresholds and elaborate on the function to selected the maximally allowed action (line 10– 12).

**Exploration.** The RL agent operates on layer-based states $s^{(l)}$ and predicts a compression rate $a^{(l)}$. We then order the values by magnitude (OWM) and introduce a threshold $\psi_K^{(l)}$ that implements the determined compression rate $a^{(l)}$. Values lower than $\psi_K^{(l)}$ are zeroed out to construct the binary pruning mask $M$ (line 13). We evaluate the robustness as well as benign accuracy of the pruned network on the validation dataset to determine the agent's reward $\mathcal{R}$ (line 16) and distribute it to all state vectors. Additionally, these are stored in the replay buffer to facilitate more stable reinforcement learning.

**Reward function.** Next to the accuracy on clean, benign data $Acc_{ben}$, we additionally incorporate the adversarial robustness as "adversarial accuracy" $Acc_{adv}$ (adversarial examples that are still classified correctly) in the reward function to yield an optimal trade-off between both:

$$\mathcal{R} = Acc_{ben} + Acc_{adv}$$

For effective and fast exploration, the reward is obtained on the validation dataset only, which is sampled homogeneously from each class of the training data. For CIFAR-10, as an example, we choose 500 images from every class, such that we yield an overall number of 5,000 samples for our validation dataset and thus 10 % of the training dataset.

## 4 Evaluation

We evaluate the performance of HERACLES's non-uniform compression strategies by enhancing state-of-the-art robust-aware pruning methods (Section 4.1), before we analyze the found strategies (Section 4.2) and discuss our method's convergence (Section 4.3). For this, we experiment with multiple architectures that are adversarially pre-trained on different datasets (CIFAR-10, SVHN, and ImageNet). The pruning methods then attempt to maintain accuracy *and* robustness whilst achieving high compression rates of either channels or weights.

In the following, we use CIFAR-10 as the representative for small-scale datasets and report corresponding results for SVHN in the appendix, for which the class-wise imbalance (see Table 5) makes pruning even more challenging. For both small-scale datasets, we consider ResNet18 (He et al., 2016), VGG16 (Simonyan and Zisserman, 2015) and WRN-28-4 (Zagoruyko and Komodakis, 2016), and thereby align with the experiments in related work. Since the approaches we compare to all use slightly different variants of VGG16, we settle for the definition of Sehwag et al. (2020) for all our experiments. For strategy search, we bootstrap the pruning stage with $N_{wup}$ = 100 episodes (warm-up), and $N_{srch}$ = 300 episodes for Deep-RL exploration-exploitation. Further details on the experimental setup are provided in Appendix A.2.

**Considered Adversaries.** We use PGD adversarial training for pre-training and fine-tuning, and also HERACLES's RL agent uses PGD adversarial examples (Madry et al., 2018) to validate the pruned network during strategy search. To generate these, we initialize with random noise and make 10 perturbation steps per sample. For CIFAR-10 and SVHN, models are trained with the maximal $l_\infty$ perturbation budget and step sizes of $\frac{8}{255}$ and $\frac{2}{255}$, respectively. For ImageNet, we use "free adversarial training" (Shafahi et al., 2019) with 4 replays, where the perturbation parameters are set to $\frac{4}{255}$ and $\frac{1}{255}$. The robustness (accuracy on adversarial examples) of the pruned models is then evaluated with multiple attack strategies each applied to the entire testing dataset with the same perturbation strength considered during training: FGSM (Goodfellow et al., 2015), PGD-10 and PGD-20 (Madry et al., 2018), and C&W$_\infty$ (Carlini and Wagner, 2017) optimized by PGD (20 steps).

**CO$_2$ Emission.** We have conducted all our experiments on Nvidia RTX-3090 GPU cards and have consumed about 960 GPU hours in total. This amounts to an estimated total CO$_2$ emissions of 204.96 kgCO2eq when using Google Cloud Platform in region europe-west3[1].

---

[1]Calculated using the "Machine Learning Impact Calculator" at https://mlco2.github.io/impact/

### 4.1 Improving related work using Heracles

We consider two approaches, Hydra and Robust-ADMM, that use uniform compression strategies for pruning neural networks, whilst maintaining both benign accuracy ($Acc_{ben}$) and adversarial robustness, that is, the accuracy on adversarially modified inputs ($Acc_{adv}$). In the following, we show that it is possible to learn a non-uniform compression strategy that improves adversarial robustness when applied to Hydra or Robust-ADMM. Moreover, Heracles is applicable to channel and weight pruning likewise—channel pruning yields a larger potential for improvement, while weight pruning is on par with related work. We simply replace the uniformly used compression ratio of Hydra and Robust-ADMM with the strategy found by our method and present the results in Tables 2a and 2b for channel and weight pruning, respectively.

| | Method | Rate | Benign Data | FGSM | PGD-10 | PGD-20 | C&W$_\infty$ |
|---|---|---|---|---|---|---|---|
| **VGG16** | Hydra | 0.50 | 69.92 / **76.82**±0.61 | 44.63 / **51.63**±0.27 | 39.82 / **47.06**±0.43 | 39.02 / **45.96**±0.45 | 36.95 / **43.98**±0.40 |
| | | 0.10 | 10.00 / **59.83**±0.78 | 10.00 / **41.20**±0.52 | 10.00 / **38.70**±0.38 | 10.00 / **38.20**±0.61 | 10.00 / **35.37**±0.40 |
| | R-ADMM | 0.50 | 72.65 / **77.59**±0.46 | 49.52 / **51.71**±0.24 | 43.60 / **46.16**±0.16 | 43.12 / **45.45**±0.39 | 41.85 / **43.54**±0.36 |
| | | 0.10 | 56.24 / **67.04**±0.54 | 35.96 / **44.68**±0.47 | 32.65 / **41.38**±0.45 | 30.21 / **40.79**±0.47 | 28.20 / **37.98**±0.36 |
| **ResNet18** | Hydra | 0.50 | 70.36 / **77.56**±0.31 | 48.63 / **51.50**±0.32 | 42.43 / **47.14**±0.14 | 41.73 / **46.22**±0.14 | 39.27 / **44.81**±0.04 |
| | | 0.10 | 10.00 / **67.52**±0.67 | 10.00 / **44.13**±0.72 | 10.00 / **40.83**±0.58 | 10.00 / **40.27**±0.60 | 10.00 / **38.07**±0.56 |
| | R-ADMM | 0.50 | 76.99 / **78.06**±0.32 | 49.21 / **50.96**±0.24 | 44.40 / **46.11**±0.29 | 42.67 / **45.19**±0.24 | 40.51 / **44.20**±0.29 |
| | | 0.10 | 63.05 / **69.17**±0.94 | 41.94 / **44.96**±0.90 | 37.79 / **41.71**±0.81 | 36.96 / **40.67**±1.09 | 35.16 / **38.74**±0.79 |
| **WRN-28-4** | Hydra | 0.50 | 75.60 / **80.30**±0.55 | 48.25 / **53.59**±0.12 | 42.93 / **48.28**±0.27 | 41.82 / **47.23**±0.35 | 39.96 / **45.99**±0.18 |
| | | 0.10 | 10.00 / **61.93**±0.88 | 10.00 / **40.13**±0.62 | 10.00 / **37.70**±0.54 | 10.00 / **37.26**±0.55 | 10.00 / **35.20**±0.58 |
| | R-ADMM | 0.50 | **79.67** / **80.11**±0.32 | 51.58 / **53.28**±0.49 | 46.41 / **48.26**±0.55 | 45.41 / **47.35**±0.48 | 43.70 / **45.69**±0.56 |
| | | 0.10 | 66.41 / **66.68**±0.82 | 40.86 / **43.51**±0.38 | 37.59 / **40.56**±0.29 | 36.65 / **40.06**±0.32 | 34.82 / **37.70**±0.34 |

(a) Channel Pruning

| | Method | Rate | Benign Data | FGSM | PGD-10 | PGD-20 | C&W$_\infty$ |
|---|---|---|---|---|---|---|---|
| **VGG16** | Hydra | 0.10 | 77.51 / **78.21**±0.60 | 50.70 / **52.00**±0.21 | 45.66 / **47.11**±0.26 | 44.57 / **46.07**±0.36 | 43.24 / **44.68**±0.23 |
| | | 0.01 | **63.99** / **64.33**±0.71 | 42.28 / **44.58**±0.66 | 39.40 / **41.44**±0.55 | 38.86 / **40.85**±0.57 | 36.34 / **38.66**±1.00 |
| | R-ADMM | 0.10 | **75.41** / 74.38±5.57 | **49.18** / 49.08±4.05 | 44.37 / **45.31**±2.76 | **43.33** / 43.70±4.18 | 41.28 / **43.35**±2.80 |
| | | 0.01 | 47.40 / **49.85**±1.93 | 35.22 / **35.30**±1.21 | 31.10 / **33.02**±1.10 | 30.85 / **32.58**±1.12 | 30.24 / **32.66**±1.94 |
| **ResNet18** | Hydra | 0.10 | 78.14 / **78.58**±0.58 | 51.14 / **51.45**±0.35 | **46.75** / 46.45±0.27 | **45.90** / 45.48±0.36 | 43.66 / **44.42**±0.24 |
| | | 0.01 | 71.01 / **72.92**±0.96 | 47.53 / **48.42**±0.62 | 43.42 / **44.43**±0.58 | 42.37 / **43.69**±0.55 | 40.25 / **41.94**±0.60 |
| | R-ADMM | 0.10 | 78.33 / **78.60**±0.79 | 50.68 / **50.91**±0.69 | 45.46 / **45.87**±0.78 | 44.48 / **44.85**±0.88 | 43.23 / **43.57**±0.62 |
| | | 0.01 | **67.01** / 62.91±1.90 | 43.57 / **43.99**±1.30 | 40.35 / **40.80**±1.12 | 39.71 / **40.26**±1.08 | 37.58 / **41.38**±1.78 |
| **WRN-28-4** | Hydra | 0.10 | **81.91** / 81.77±0.38 | **54.26** / 53.86±0.49 | 47.87 / **47.82**±0.68 | 46.80 / **46.77**±0.78 | 46.07 / **45.86**±0.66 |
| | | 0.01 | 68.74 / **71.63**±0.72 | 44.83 / **46.65**±0.64 | 41.10 / **43.36**±0.46 | 40.12 / **42.70**±0.51 | 38.95 / **40.71**±0.58 |
| | R-ADMM | 0.10 | 80.10 / **80.12**±0.57 | 51.98 / **52.73**±0.48 | 46.98 / **47.34**±0.78 | 46.03 / **46.45**±0.89 | 44.43 / **45.03**±0.65 |
| | | 0.01 | 59.88 / **61.68**±0.82 | 38.31 / **39.74**±0.65 | 36.28 / **37.09**±0.63 | 35.98 / **36.61**±0.68 | 35.76 / **36.83**±0.79 |

(b) Weight Pruning

Table 2: Uniform vs. non-uniform pruning on CIFAR-10 with Hydra and Robust-ADMM. The accuracy of both strategies is presented in [%] left and right of the / character, respectively, considering benign input data and 4 different attacks. Non-uniform strategies generated by Heracles are averaged over 5 experiments and show the standard deviation in ± notation.

**CIFAR-10.** Heracles's compression strategy can improve the performance of VGG16, ResNet18, and WRN-28-4 pruned by Hydra as well as Robust-ADMM. As an example, for pruning channels

at a compression rate of $a_c = 0.5$, the benign and adversarial accuracy in VGG16 pruned by Hydra increase 6.90 and up to 7.03 (C&W$_\infty$) percentage points, respectively. A similar trend is observed for Robust-ADMM and other architectures as well, with WRN-28-4 being the most challenging setting. For aggressive channel pruning ($a_c = 0.1$), Heracles enables Hydra to even avoid an completely damaged model that yields 10 % accuracy and thus random outputs. For weight pruning, the results are less obvious. While Heracles's non-uniform strategies do not yield similar high levels of improvement they are still slightly better or on par with the uniform compression rates. Again, the results are consistent across architectures for both, Hydra and Robust-ADMM.

**SVHN**. As a second small-sized dataset, we have conducted experiments on the SVHN datasets. While similar in size to the CIFAR-10 dataset, SVHN is highly unbalanced (as shown in Table 5), which can pose additional challenges. In Appendix A.3, we report details on the results and visualize the compression strategies found by Heracles for the different architectures in Fig. 4.

**ImageNet**. Next, we apply our method for moderate pruning of ResNet50 learned on the large-scale dataset ImageNet. Due to the size of the dataset, we reduce the RL agent's validation dataset for strategy search to 1 % of the training data. Sehwag et al. (2020) demonstrate Hydra's effectivity for pruning weights of a model learned on ImageNet and we are able to confirm these results in our experiment as shown in Table 3.

| Pruning | Method | Benign Data | FGSM | PGD-10 | PGD-20 | C&W$_\infty$ |
|---|---|---|---|---|---|---|
| channels ($a_c = 0.5$) | Hydra | 46.67 / **50.03** | 24.34 / **26.18** | 21.45 / **23.61** | 20.79 / **22.45** | 19.36 / **21.06** |
| | R-ADMM | 48.62 / **50.52** | 21.16 / **23.85** | 21.19 / **22.15** | 21.21 / **23.88** | 19.36 / **21.47** |
| weights ($a_w = 0.1$) | Hydra | **49.08** / 48.71 | **26.26** / 25.81 | **23.25** / 23.19 | **22.75** / 22.31 | **21.21** / 20.70 |
| | R-ADMM | 35.83 / **37.45** | 15.42 / **16.51** | 14.89 / **15.89** | 14.88 / **15.83** | 12.60 / **13.81** |

Table 3: Uniform vs. non-uniform pruning on ImageNet with Hydra and Robust-ADMM. The accuracy of both strategies is presented in % left and right of the / character, respectively.

While for Hydra results with non-uniform compression remains similar, Heracles's strategies can improve Robust-ADMM in weight pruning. However, the added value does not suffice to help Robust-ADMM surpass Hydra. Moreover, in channel pruning both Hydra and Robust-ADMM show obvious improvements over uniform compression by using our method's strategies and both methods are then nearly on-par in robustness and performance on benign data.

## 4.2 Analysis of Heracles's Strategies

We take weight pruning ($a_w = 0.1$) and channel pruning ($a_c = 0.5$) on CIFAR-10  as an example and inspect the global compression strategies learned by our method. Fig. 2 visualizes the learned strategies by Heracles for VGG16, ResNet18, and WRN-28-4.

**Channel pruning**. The learned strategies for channel pruning (orange lines) consistently preserve more parameters in the first several layers and prune the convolutional layers at the end of the networks more aggressively. The strategy, however, differs in compression rates of fully connected layers of the network architectures. For VGG16, it is notable that the RL agent preserves much more of them than the middle convolutional layers. Interestingly, in residual block based networks (ResNet18 and WRN-28-4) the RL agent discovers pruning potential on the last connected layer.

**Weight pruning**. With an overall compression rate of $a_w = 0.1$, the learned strategies for weight pruning (blue lines) are more diverse for the individual network architectures. Networks with residual blocks share parameters which causes a more homogeneous parameter distribution on each layer. As an example, for ResNet18 the agent does not preserve front layers but prunes layers

more homogeneously. Also for WRN-28-4 the pruning strategy approaches uniformity, which also explains the similarity in results between uniform and non-uniform strategies in Table 2b. For VGG16 (a conventional CNN without shortcut layers) in contrast, HERACLES particularly preserves layers in the front and prunes layers in the back more distinctively.

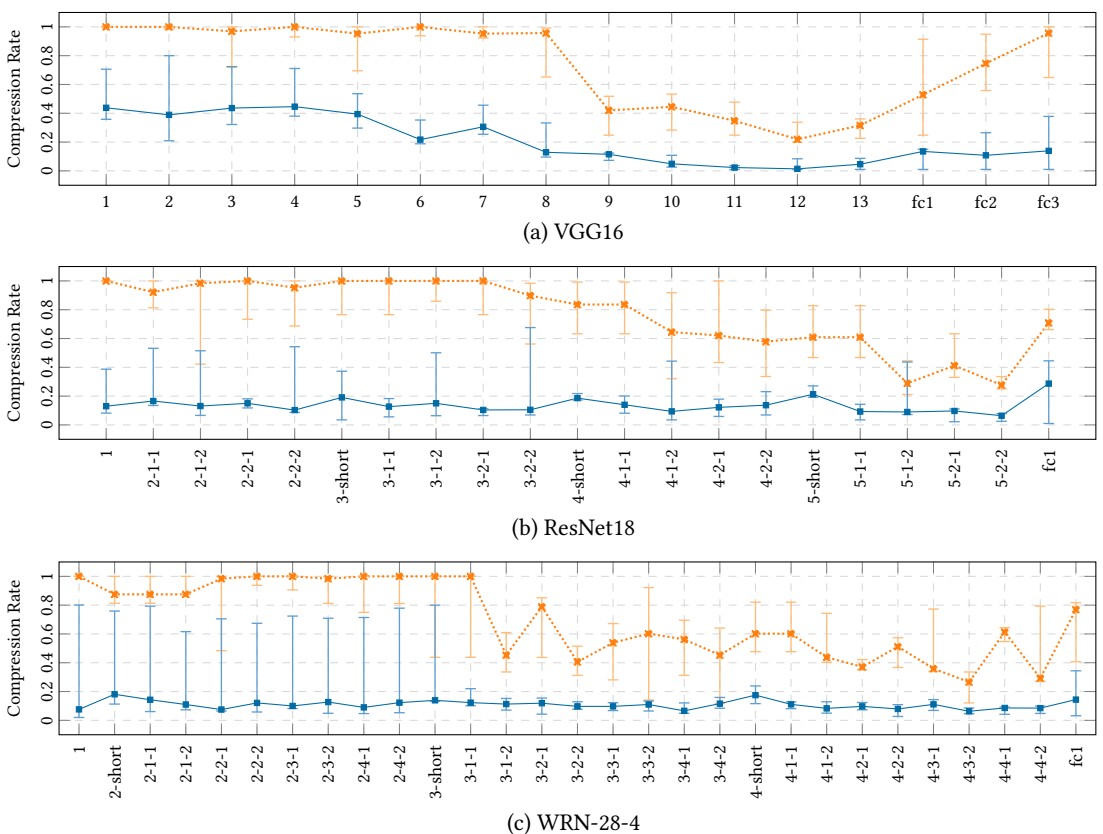

Figure 2: HERACLES's strategies for pruning channels ($a_c$ = 0.5; dashed orange line) and weights ($a_w$ = 0.1; solid blue line) of VGG16, ResNet18, and WRN-28-4 on CIFAR-10.

## 4.3 Pruning Convergence

We have shown the capability of HERACLES's strategies to outperform related work for compressing channels on CIFAR-10, but our evaluation also exposes the lack of significant improvement when pruning weights. In this section, we take ResNet18 as an example to inspect the RL agent's searching progress in Fig. 3 to detail the underlying reasons. Top sub-figures (a and b) refer to moderate compression, bottom ones (c and d) show very aggressive pruning. Left sub-figures (a and c) belong to channel pruning, whereas right sub-figures (b and d) show weight pruning.

**Channel pruning**. Convergence for moderate pruning at $a_c$ = 0.5 works flawlessly. After 300 steps the RL agent has successfully determined a strategy that reaches the highest reward. While high initial exploration leads to large fluctuation, after 350 episodes the reward converges. At $a_c$ = 0.1, in turn, model performance is strongly degraded by the highly aggressive pruning. However, the RL agent keeps excavating better strategies, yielding good results eventually (cf. Table 2a).

**Weight pruning**. At $a_w$ = 0.1, the process exhibits a certain instability due to the high sensitivity to the compression rate. Still, the final stage converges to the overall best reward. Differently, aggressive pruning at $a_w$ = 0.01 hinders successful exploration. The best strategies found, thus, merely realize performance on-par with uniform pruning.

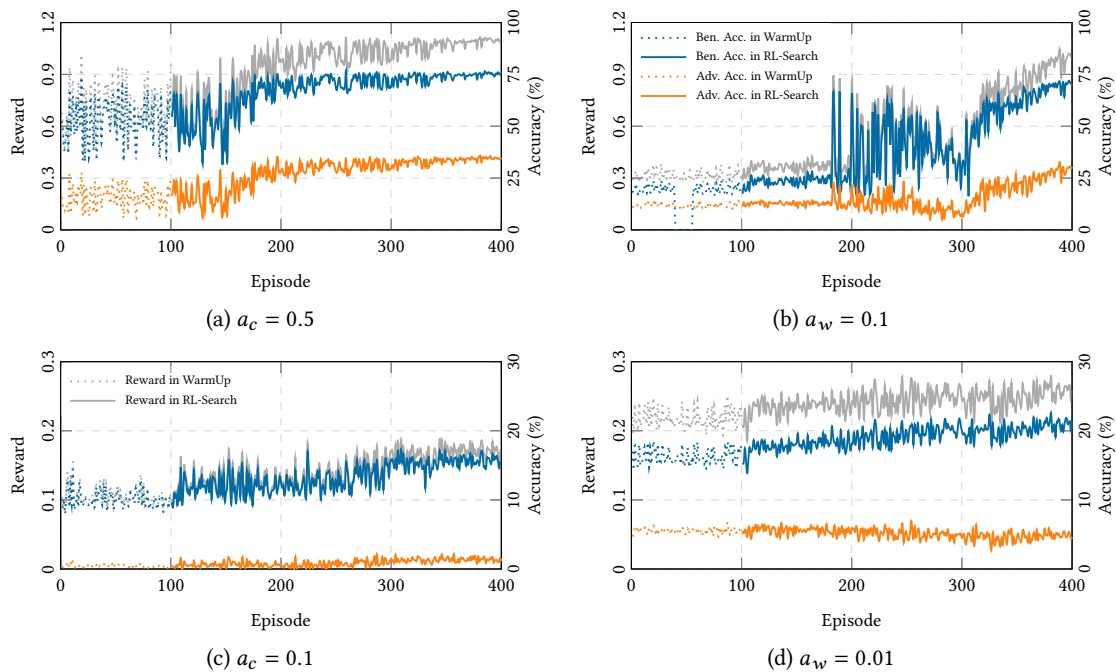

Figure 3: Convergence of HERACLES's RL agent for pruning ResNet18 on CIFAR-10.

## 5 Limitations

HERACLES exhibits three noteworthy limitations in practice: First, due to the *randomness* inherent to reinforcement learning exploration, our method exhibits variance in the yield results and ultimately cannot guarantee that the RL agent converges to an optimal strategy. Our experiments, however, show that this still succeeds in the majority of the cases, surpassing the state of the art. Second, HERACLES's non-uniform strategies are *more effective for channel pruning than weight pruning*. This can be seen directly in the reported accuracy, but also the convergence visualization in Fig. 3, where channel pruning (left) is more effective and stable than weight pruning (right)–even at aggressive compression rates. Third, one needs to pay attention to *runtime performance* and exploration efficiency. Strategy search for pruning weights of the evaluated CIFAR-10 models, for instance, requires 4.6× longer on average than channel pruning on a single NVIDIA RTX-3090 card. This further emphasizes HERACLES's primary suitability for channel pruning. Moreover, models with pruned channels will also be more resource-friendly during inference as channel pruning reduces the dimensionality of computations rather than zeroing out single values.

## 6 Conclusion

Striking a balance between benign accuracy and adversarial robustness during pruning is challenging. Related work has already shown impressive results using uniform compression strategies. With HERACLES, we present a method that learns a global but layer-specific and thus *non-uniform* compression strategy, which can be used to further benefit existing, state-of-the-art approaches. For instance, we increase performance for aggressive channel-pruning ($a_c = 0.1$) with Robust-ADMM by up to 10.80 and 9.78 (C&W$_\infty$) percentage points for benign and adversarial accuracy, respectively. Weight pruning using our compression strategies has shown less distinctive results but still is slightly better or on par with related work. This is founded in the fact that here the best compression strategy often is close to uniformity. If so HERACLES also finds these close-to-uniform strategies. Such flexibility when pruning deep neural networks is crucial in practice to adapt to the model at hand. The results using non-uniform strategies are particularly promising on channel granularity, where HERACLES significantly improves related work on adversarially robust pruning.

## Acknowledgments

The authors thank the anonymous reviewers for their valuable suggestions, and gratefully acknowledge funding by the Helmholtz Association (HGF) within topic "46.23 Engineering Secure Systems" and by SAP S.E. under project DE-2020-021.

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

## A Appendix

In the appendix, we extend on three aspects of our method: We begin by detailing the process of layer-wise action rate control to reach the preset target compression rate. After that, we further describe our experimental setup, before we investigate the performance of HERACLES's non-uniform strategy on SVHN.

### A.1 Action control

The action control algorithm helps to constrain the layer's compression rate to reach the targeted global network size. In related work (Boyd et al., 2011; Sehwag et al., 2020), the global compression rate is defined as the sparsity rather than the ratio of preserved parameters for both weight and channel pruning. Weight pruning is apt to be controlled by the sparsity. However, since sparsity on the channel granularity can not reflect the real network size after pruning with a global non-uniform strategy, sparsity control is not appropriate anymore. Our approach, thus, considers the influence from layer $l$ on its preceding (joint) layer $l - 1$ for channel pruning, such that the pruned network has the exact same size as considered in related work. We denote determining the global parameter-wise compression rate as the function *ConvertRate* in Algorithm 2.

---

**Algorithm 2** Max-Allow-Action

---

**Input**: Objective layer $l$, Number of total network parameters $\Theta_{all}$, Number of network layers $L$, Target rate $a_t$, Rate range $[a_{min}, a_{max}]$, Founded action list $[a^1, \ldots, a^{l-1}]$, Pruning regularity *P-reg*
**Output**: Maximal allowed action $a_{allow}^{(l)}$

*P-reg = "Weight Pruning"*:
1: Convert compress rate:
   $\quad a_t = ConvertRate(a_t, \text{P-reg})$

2: Initialize: $a^{(l)} = 1.0$
3: $\Theta_t = a_t \cdot \Theta_{all}$

4: $\widetilde{v} = \sum_{i=1}^{l-1} a^{(i)} \cdot \Theta^{(i)}$
5: $v = \sum_{i=l+1}^{L} a_{min} \cdot \Theta^{(i)}$

6: $Duty = \Theta_t - (\widetilde{v} + v)$

7: $a_{allow} = \frac{Duty}{\Theta^{(l)}}$

8: $a_{allow}^{(l)} = \min(a_{allow}^{(l)}, a_{max})$

*P-reg = "Channel Pruning"*:
1: Convert compress rate:
   $\quad a_t = ConvertRate(a_t, \text{P-reg})$

2: Initialize: $a^{(l)} = 1.0$
3: $\Theta_t = a_t \cdot \Theta_{all}$

4: $\widetilde{v} = \sum_{i=1}^{l-2} a^{(i)} \cdot a^{(i+1)} \cdot \Theta^{(i)}$
5: $v = \sum_{i=l+1}^{L-1} a_{min}^2 \cdot \Theta^{(i)} + a_{min} \cdot \Theta^{(L)}$

6: $Duty = \Theta_t - (\widetilde{v} + v)$

7: $a_{allow}^{(l)} = \frac{Duty}{a_{min} \cdot \Theta^{(l)} + a^{(l-1)} \cdot \Theta^{(l-1)}}$

8: $a_{allow}^{(l)} = \min(a_{allow}^{(l)}, a_{max})$

---

### A.2 Experimental setup

We conduct experiments on three different datasets: CIFAR-10, SVHN, and ImageNet. In the following, we elaborate on the considered networks and their pre-trained performances as used in our experiment, the action range used by the action-control algorithm, and settings of the RL agent doing the strategy search.

**Networks.** In related work, a variety of different deep neural network architectures are used for evaluating pruning approaches. Some of the supposedly identical networks, however, show subtle differences. For a fair comparison, we thus center our experiments on the architectures used by Sehwag et al. (2020) and build the exact same VGG16, ResNet18, and WRN-28-4 as in their open-source implementation[2]. For our experiments on the large-scale ImageNet dataset, we additionally use ResNet50 as proposed by He et al. (2016). We conduct 90 and 100 epochs for pre-training models on the large-scale and the small-scale datasets, respectively, with a learning rate starting

---

[2]`https://github.com/inspire-group/hydra`

at 0.1 while adapting it with a cosine learning-rate schedule (Loshchilov and Hutter, 2016). For pruning with HERACLES's strategies, we employ the respective default schedules in both HYDRA and Robust-ADMM to adapt the learning rate. Table 4 summarizes the used network architectures and lists their performance after adversarial pre-training. These models are used for all pruning experiments to ensure an identical starting point for our comparison.

| Model/ Network Architecture | | CIFAR-10 | SVHN | ImageNet |
|---|---|---|---|---|
| VGG16 | as used by Sehwag et al. (2020) | 75.72 / 46.35 | 92.40 / 55.09 | — |
| ResNet18 | as proposed by He et al. (2016) | 82.68 / 43.44 | 92.70 / 59.33 | — |
| WRN-28-4 | as proposed by Zagoruyko and Komodakis (2016) | 83.35 / 48.86 | 92.69 / 57.15 | — |
| ResNet50 | as proposed by He et al. (2016) | — | — | 60.25 / 32.82 |

Table 4: The network architectures used in this work and their accuracy after adversarial pre-training for benign data and PGD-10 attacks, left and right of the / character for different dataset.

**Action range**. The range of compression rates $(a_{min}, a_{max})$ considered by the RL agent (the action values) have to be specified upfront, for which a few things need to be followed: For aggressive pruning, sufficient neurons must remain in each layer rather than being pruned entirely. For moderate pruning, the agent has to be encouraged to explore different possibilities. Consequently, we set the range as $[0.01, 0.8]$ and $[0.005, 0.5]$, for weight pruning with compression rates of $a_w = 0.1$ and $a_w = 0.01$, respectively. For channel pruning with compression rates of $a_c = 0.5$ and $a_c = 0.1$, in turn, we use $[0.1, 1.0]$ and $[0.05, 0.5]$. Note that for channel pruning, we maintain a compression rate of 1.0 (no compression) for the first layer to keep input information intact. Moreover, for residual blocks, we set the compression rate on the shortcut layers to the same value as for the connected backbone layers, such that networks with residual blocks are processable by the channel-wise pruning strategy.

**RL agent setting**. We use DDPG (Lillicrap et al., 2016) as RL agent for determining the layer's state and, thus, predict compression rates. In our implementation, the actor network and the critic network are both constructed with two 300 neurons wide, fully connected layers. Moreover, the size of the replay buffer is set to 200 times the number of prunable layers in the neural network—for instance, for pruning weights of VGG16 the size equals to $200 \times 16 = 3,200$. The training of the RL agent is performed with learning rates 0.01 and 0.001 respectively on critic and actor in DDPG. And we use a soft update of 0.01 on the target model. During the 300 episodes long RL search phase, we train the agent for 20 epochs on states sampled from the replay buffer with a batch size of 128. For better exploration, we additionally set the exponential decay $\delta$ to 0.99.

### A.3 HERACLES's performance on SVHN

In contrast to CIFAR-10, neither training nor testing data of SVHN is balanced. While we use the common accuracy to determine robustness and the model's natural performance in our experiments on CIFAR-10, for SVHN we hence use the balanced accuracy (Brodersen et al., 2010) as the evaluation metric. Table 5 shows the class-wise distribution of the SVHN dataset. For both datasets, a completely damaged model with random predictions thus is indicated by a (balanced) accuracy of 10 % as the datasets contain 10 classes each.

**Channel pruning**. Table 6a summarizes the experimental results of channel pruning on SVHN. With the help of HERACLES, both HYDRA and Robust-ADMM improve robustness and natural accuracy at $a_c = 0.5$. Interestingly, pruning ResNet18 by Robust-ADMM equipped with our strategies even yields up to 9.05 percentage points higher robustness against PGD-10 than uniform compression. However, the variance is rather high, which originates the inherent randomness of reinforcement

| Dataset | Amount of Samples per Class [%] | | | | | | | | | |
|---|---|---|---|---|---|---|---|---|---|---|
| | 1 | 2 | 3 | 4 | 5 | 6 | 7 | 8 | 9 | 10 |
| Training | 18.92 | 14.45 | 11.60 | 10.18 | 9.39 | 7.82 | 7.64 | 6.89 | 6.36 | 6.75 |
| Test | 19.59 | 15.94 | 11.07 | 9.69 | 9.16 | 7.59 | 7.76 | 6.38 | 6.13 | 6.70 |

Table 5: Class-wise data distribution of the SVHN dataset.

learning. The accuracy on benign data, in turn, remains stable across all strategies and our method yields a pruned model with at least a performance on-par with the uniform strategy. This clearly shows the positive impact of HERACLES in moderate pruning. Aggressive channel pruning ($a_c = 0.1$) is more challenging for SVHN, though. Neither uniform nor our non-uniform compression can

| | Method | Rate | Benign Data | FGSM | PGD-10 | PGD-20 | C&W$_\infty$ |
|---|---|---|---|---|---|---|---|
| ResNet18 | Hydra | 0.50 | **91.05** / **91.52**±0.63 | 65.17 / **67.06**±3.96 | 53.76 / **54.16**±1.62 | 51.28 / **51.74**±0.90 | 48.63 / **49.66**±1.52 |
| | | 0.10 | 10.00 / **82.74**±0.56 | 10.00 / **46.09**±1.38 | 10.00 / **43.31**±0.41 | 10.00 / **38.87**±1.02 | 10.00 / **34.95**±0.76 |
| | R-ADMM | 0.50 | **91.59** / **91.94**±0.38 | 74.59 / **80.45**±6.18 | 56.95 / **66.00**±8.67 | 54.06 / **60.42**±6.79 | 51.92 / **59.89**±7.22 |
| | | 0.10 | **86.65** / 85.96±1.46 | 56.43 / **59.16**±1.53 | **47.05** / 46.52±0.36 | **45.34** / 44.72±0.10 | 41.47 / **41.54**±0.25 |
| VGG16 | Hydra | 0.50 | 89.27 / **91.18**±0.32 | 59.09 / **62.01**±0.34 | 50.07 / **52.14**±0.35 | 48.01 / **49.42**±0.22 | 44.06 / **46.52**±0.17 |
| | | 0.10 | 10.00 / 10.00±0.00 | 10.00 / 10.00±0.00 | 10.00 / 10.00±0.00 | 10.00 / 10.00±0.00 | 10.00 / 10.00±0.00 |
| | R-ADMM | 0.50 | 88.72 / **89.97**±0.97 | 57.02 / **60.63**±1.20 | 48.15 / **51.09**±0.95 | 46.03 / **49.20**±0.94 | 42.72 / **46.33**±0.56 |
| | | 0.10 | 79.25 / **85.27**±2.77 | 45.46 / **51.54**±2.74 | 37.92 / **43.12**±2.23 | 36.39 / **41.28**±2.15 | 32.39 / **37.22**±2.36 |
| WRN-28-4 | Hydra | 0.50 | **91.37** / 91.32±0.38 | 65.01 / **67.42**±2.27 | 54.84 / **55.64**±1.03 | **53.31** / 53.74±0.45 | 50.16 / **51.10**±0.96 |
| | | 0.10 | 10.00 / **81.30**±0.86 | 10.00 / **48.22**±0.98 | 10.00 / **41.32**±0.96 | 10.00 / **39.94**±0.95 | 10.00 / **36.22**±0.97 |
| | R-ADMM | 0.50 | **91.57** / **92.03**±1.08 | 73.49 / **74.35**±2.41 | 57.48 / **61.45**±8.49 | 54.85 / **58.04**±6.29 | 53.47 / **56.62**±7.17 |
| | | 0.10 | **89.22** / 88.94±0.48 | **57.77** / 50.21±1.42 | **47.55** / 47.37±0.76 | **45.60** / 40.83±1.93 | **42.40** / 37.26±1.52 |

(a) Channel Pruning

| | Method | Rate | Benign Data | FGSM | PGD-10 | PGD-20 | C&W$_\infty$ |
|---|---|---|---|---|---|---|---|
| ResNet18 | Hydra | 0.10 | **91.36** / **91.61**±0.13 | 64.04 / **66.95**±1.94 | 52.74 / **53.61**±0.71 | 50.63 / **51.31**±0.49 | 48.14 / **49.07**±0.52 |
| | | 0.01 | **88.23** / 88.21±0.21 | 58.76 / **59.72**±0.81 | 49.80 / **50.62**±0.24 | 48.05 / **49.07**±0.19 | 44.86 / **45.34**±0.23 |
| | R-ADMM | 0.10 | 89.73 / **91.71**±0.88 | 77.39 / **79.11**±7.46 | 57.01 / **61.69**±6.34 | 55.25 / **56.60**±3.78 | 53.35 / **55.16**±4.83 |
| | | 0.01 | 84.63 / **85.11**±1.51 | 51.10 / **55.93**±4.28 | 42.81 / **45.92**±1.71 | 40.93 / **43.38**±1.86 | 36.96 / **40.14**±2.16 |
| VGG16 | Hydra | 0.10 | 90.46 / **91.40**±0.40 | 59.87 / **62.79**±0.90 | 50.02 / **51.38**±0.66 | 47.89 / **49.01**±0.63 | 44.84 / **46.39**±0.51 |
| | | 0.01 | 84.68 / **89.30**±0.53 | 52.37 / **58.90**±0.82 | 45.34 / **50.40**±0.56 | 43.60 / **48.35**±0.59 | 39.16 / **44.89**±0.66 |
| | R-ADMM | 0.10 | 89.12 / **90.10**±0.33 | 58.61 / **59.73**±0.43 | 48.62 / **51.25**±0.32 | 46.56 / **48.66**±0.36 | 43.78 / **45.63**±0.35 |
| | | 0.01 | 55.80 / **86.46**±2.38 | 28.55 / **55.32**±2.13 | 24.34 / **47.58**±0.67 | 23.48 / **45.70**±1.13 | 21.15 / **41.71**±0.72 |
| WRN-28-4 | Hydra | 0.10 | 91.61 / **92.16**±0.15 | **67.72** / 67.52±2.23 | 54.76 / **55.06**±0.88 | 52.74 / **52.87**±0.45 | **50.31** / 50.16±0.90 |
| | | 0.01 | **88.37** / 82.85±4.93 | **57.66** / 54.45±2.82 | **49.24** / 46.42±2.27 | **46.34** / 45.12±2.19 | **43.97** / 42.58±0.93 |
| | R-ADMM | 0.10 | **90.36** / **90.78**±0.57 | **68.33** / 65.83±1.62 | **53.92** / 53.41±0.72 | **52.19** / 50.71±0.95 | **49.66** / 48.58±0.66 |
| | | 0.01 | 10.00 / 10.00±0.00 | 10.00 / 10.00±0.00 | 10.00 / 10.00±0.00 | 10.00 / 10.00±0.00 | 10.00 / 10.00±0.00 |

(b) Weight Pruning

Table 6: Uniform vs. non-uniform pruning on SVHN with HYDRA and Robust-ADMM. The balanced accuracy of both strategies is presented in % left and right of the / character, respectively, considering benign input data and 4 different attacks. Non-uniform strategies generated by HERACLES are averaged over 5 experiments and show the standard deviation in ± notation.

prevent VGG16 from complete damage (10 %) and the uniform strategy adapts better to WRN-28-4 when pruning with Robust-ADMM. In all other situations, Heracles's strategies still yield on-par or better performance in aggressive channel pruning as well. Even in situations where uniform compression yields a completely damaged model (ResNet18 and WRN-28-4 using Hydra), our method selects a strategy that results in meaningful and competitive results.

Furthermore, we visualize the strategies learned by Heracles for moderate channel pruning in Fig. 4 (orange lines). Similar to our results on CIFAR-10, the RL agent discovers high redundancy (and thus pruning potential) in the middle convolutional layers for VGG16. Differently, the compression on ResNet18 is nearly uniform along convolutional layers where the RL agent, however, still acknowledges the higher importance of the final fully connected layer.

**Weight pruning**. In comparison to channel pruning, Heracles is more stable and achieves higher robustness on VGG16 and ResNet18 (cf. Table 6b). For both moderate ($a_w = 0.1$) and aggressive ($a_w = 0.01$) pruning, state-of-the-art methods can benefit from Heracles's strategies and improves up to 30.66 and 26.77 percentage points for benign accuracy and FGSM robustness, respectively. However, also here similar instability issues happen for pruning ResNet18 with Robust-ADMM. In Fig. 4b, we see that the RL agent approaches a strategy close to uniformity for WRN-28-4 and ResNet18. For the latter, in particular for early layers the variance is larger than for layers further back. Overall, we observe that non-uniform compression is more beneficial for channel pruning than weight pruning. However, for aggressive weight pruning ($a_w = 0.01$) by Robust-ADMM, WRN-28-4's performance is completely damaged for uniform and non-uniform strategies alike.

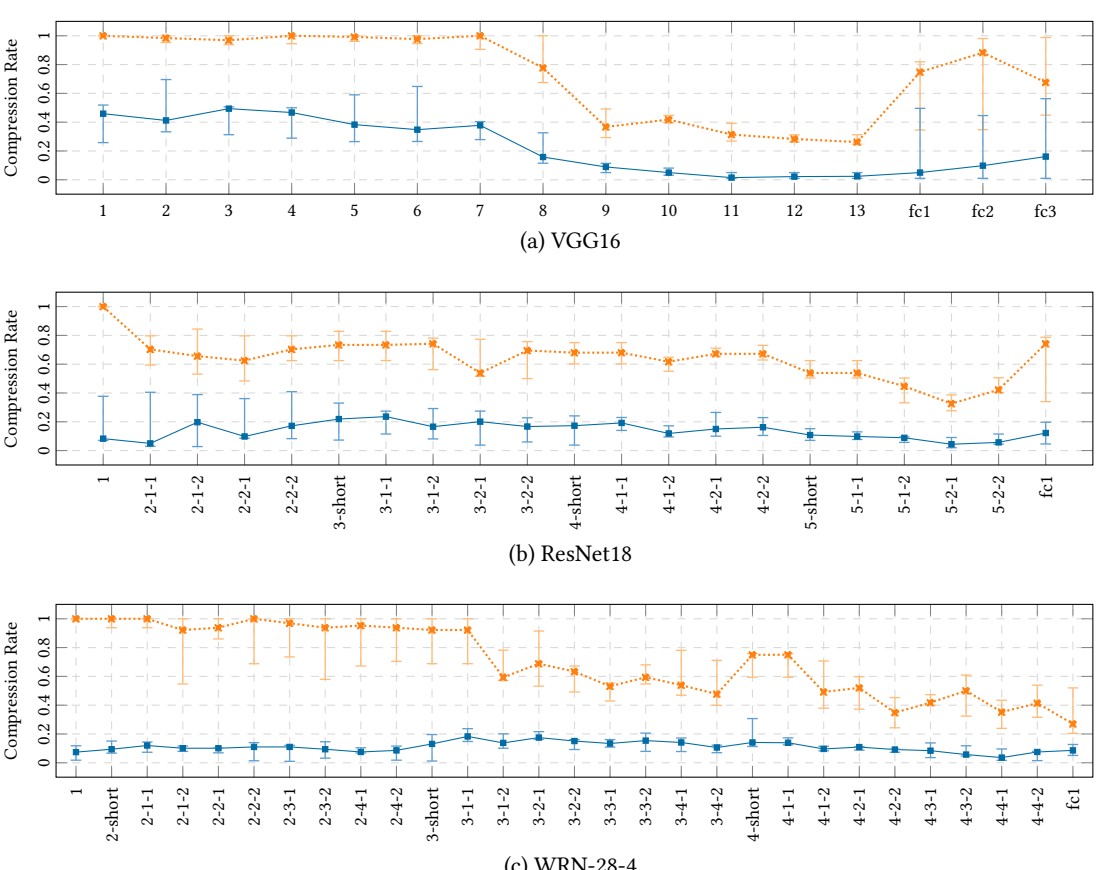

(a) VGG16

(b) ResNet18

(c) WRN-28-4

Figure 4: Heracles's strategies for pruning channels ($a_c = 0.5$; dashed orange line) and weights ($a_w = 0.1$; solid blue line) of VGG16, ResNet18, and WRN-28-4 on SVHN.

