# OpenReview forum: "Non-Uniform Adversarially Robust Pruning"
_automl.cc/AutoML/2022/Track/Main — AutoML-Conf 2022 (Main Track)_

### Official Review · Reviewer_29Xr · 2022-03-28

**Potential Impact On The Field Of Automl Rating:** 2
**Technical Quality And Correctness:** 1. The developed RL-based searching a…
**Technical Quality And Correctness Rating:** 4
**Clarity Rating:** 3

**Summary Of Contributions:**

This paper shows that employing non-uniform compression strategies allows conjoining model compactness and adversarial robustness. To realize this, the paper develops an RL algorithm to search for layer-specific, non-uniform compression ratios. The empirical studies are thorough and convincing. This paper is problem-oriented, and the technical novelty, honestly, is limited.


**Clarity:**

The equation between L157 and L158 is informal.

Is there any evidence showing the training consumption of the RL agent?

**Overall Review:**

- Positive aspects:
1. This paper leverages AutoML strategies (i.e., RL-based hyper-parameter search) to solve the network pruning problem.
2. The paper may bring insights and exhibit impact on the network pruning community.

- Negative aspects:
1. The technical novelty is limited.
2. RL-based searching may be inefficient at scale.

**Potential Impact On The Field Of Automl:**

This paper leverages AutoML strategies (i.e., RL-based hyper-parameter search) to solve the network pruning problem. Thus, the paper may bring insights and exhibit impact on the network pruning community, but I am not sure whether the finding of the value of non-uniform pruning is novel or not.

**Reproducibility:**

The results may be reproducible.

**Review Confidence:**

3: You are fairly confident in your assessment. It is possible that you did not understand some parts of the submission or that you are unfamiliar with some pieces of related work.

**Review Rating:**

4: Marginally above the acceptance threshold (use sparsely)

**Review Summary:**

The paper solves a seemingly important problem with well-studied AutoML strategies, so I am learning to accept this paper but my confidence is low.

---

### Official Review · Reviewer_PW9A · 2022-04-01

**Potential Impact On The Field Of Automl Rating:** 3
**Technical Quality And Correctness Rating:** 2
**Clarity Rating:** 3
**Ethics Rating:** Yes, other reasons (please specify be…

**Summary Of Contributions:**

The paper claims that a *non-uniform compression* improves the final accuracy of the pruned models on the original data and adversarially perturbed data. In this regard, the authors propose a Reinforcement-Learning-based method to search for a non-uniform pruning strategy that will replace the uniform compression ratios used in other state-of-the-art methods afterward. For each layer, the RL agent estimates the compression ratio to use following the Order of Weight Magnitude criterion such that the global compression rate is constraint is satisfied. The reward for the agent is computed by evaluating the benign and robust accuracies of the pruned model obtained to reach an optimal trade-off between both.
The strategy found is implemented on two state-of-the-art robust pruning method, HYDRA and Robust-ADMM, and evaluated on the CIFAR-10 dataset in the main paper and SVHN and ImageNet datasets in the Appendix. Two types of pruning are compared, channel and weight pruning, for multiple architectures.

**Clarity:**

I had a hard time understanding the paper at first read. I'm still not sure to understand the specific difference between channel pruning and weight pruning, and since I did not find other references to channel pruning in the literature, it makes it difficult to assess the results obtained in this setting. I think that a figure describing the full experimental process and illustrating the algorithm would improve the clarity of the paper.

**Ethics Details (Optional):**

The paper does not include a discussion about the broader impact of their work, even though it was required in the call.

**Overall Review:**

### Positive

- The proposed idea of using a non-uniform compression rate is simple and intuitive. Indeed, some parts of the network might be more important than others. As the experiments has shown in the paper, the first layers of convolutional networks seem to be more important to keep.
- When channel pruning, using a non-uniform strategy clearly improves the performance and robustness of the different prunning algorithm used. It can also prevent the algorithm from completely damaging the model when pruning uniformly.
- The experiments are done on multiple datasets (CIFAR-10, SVHN and ImageNet), for different global compression rates and using several architectures.

### Negative

- The most important experiments, those on the ImageNet dataset, are relegated to the Appendix when they should clearly appear in the main paper since it corresponds to the most difficult setting. Only the experiments on CIFAR-10 are presented in the main paper.
- The importance of a non-uniform strategy is overstated in the paper. On the ImageNet dataset on the weight pruning setting, which seems to be the most important setting as it is the one considered in all previous works, the effect of a non-uniform strategy is less clear.
- The clarity of the paper could be improved with figures explaining the whole process and illustrating the algorithm to find the compression strategy to use.

**Potential Impact On The Field Of Automl:**

The idea of a non-uniform compression strategy is really interesting, and the paper achieves promising results by improving state-of-the-art methods. This idea could be implemented into future methods or leads to novel design of pruning algorithms. This paper can foster future work on this topic as well.

**Reproducibility:**

The reproducibility list was correctly filled out, the code to run the experiments was made available, and all the hyperparameters are presented in the paper.

**Review Confidence:**

3: You are fairly confident in your assessment. It is possible that you did not understand some parts of the submission or that you are unfamiliar with some pieces of related work.

**Review Rating:**

4: Marginally above the acceptance threshold (use sparsely)

**Review Summary:**

The main idea of the paper could inspire future work on non-uniform pruning algorithms, as the experiments show promising results. However, the improvement is not so clear on more difficult settings and the claim should be toned down. The clarity of the method could be improved with detailed figures.

**Technical Quality And Correctness:**

The proposed approach is sounded, and the experiments show that it globally improves over state-of-the-art *for the channel pruning case*. However, for weight pruning, the improvement is not so clear, especially for HYDRA (the best method) on the ImageNet dataset in the Appendix. As pointed out in Section 4.2, the strategy learned by the agent for weight pruning is mainly uniform for bigger architectures with shortcut layers, which undermines the claims made in the paper. I think that these claims should be tone down in light of these experiments.

---

### Official Review · Reviewer_p6Cn · 2022-04-04

**Potential Impact On The Field Of Automl Rating:** 2
**Technical Quality And Correctness Rating:** 4
**Clarity:** The motivation is clear and the paper…
**Clarity Rating:** 3

**Summary Of Contributions:**

The paper proposed an algorithm to learn a non-uniform pruning policy for compressing the deep neural network and maintaining good performance under adversarial attacks. Pruning policy is learned by reinforcement learning where each state is the parameter features of each layer, i.e., convolutional kernel sizes, and the action space is the range for compression ratio. With the learned policy, the paper shows that they can achieve consistent improvement for both clean images and perturbed images compared to the uniform pruning policy, where the compression ratio of each layer is usually set manually.

**Overall Review:**

Pros:
1) The paper is easy to follow and the motivation is clear.
2) Compared to other non-uniform robust-aware methods, the proposed pruning policy can achieve great improvement for both clean and perturbed image accuracy.

Cons:
1) I am not sure how novel is the method regarding to AutoML.
2) I think it will be important to see how is the performance when we naively combine adversarial training with existing automatically pruning methods.

**Potential Impact On The Field Of Automl:**

This paper proposed an automatic way to learn to prune the robust-aware deep neural networks. The overall problem is interesting, however, I am not sure how novel it is especially in the field of AutoML since there're many papers on automatically pruning the networks. The impact here is mainly on how to combine it with adversarial training.

**Reproducibility:**

I think the reproducibility is reasonably good.

**Review Confidence:**

3: You are fairly confident in your assessment. It is possible that you did not understand some parts of the submission or that you are unfamiliar with some pieces of related work.

**Review Rating:**

3: Marginally below the acceptance threshold (use sparsely)

**Review Summary:**

I lean to reject the paper but I would like to change my mind.

**Technical Quality And Correctness:**

Technically there are no flaws to me. However, instead of comparing to non-uniform robust-aware pruned methods, can we have some results for other auto compression methods but with adversarial training?

---

### Official Review · Reviewer_L1ca · 2022-04-10

**Potential Impact On The Field Of Automl:** N/A for reproducibility reviewers
**Potential Impact On The Field Of Automl Rating:** 1
**Technical Quality And Correctness:** N/A for reproducibility reviewers
**Technical Quality And Correctness Rating:** 1
**Clarity:** N/A for reproducibility reviewers
**Clarity Rating:** 1

**Summary Of Contributions:**

N/A for reproducibility reviewers

**Ethics Details (Optional):**

N/A for reproducibility reviewers

**Overall Review:**

N/A for reproducibility reviewers

**Reproducibility:**

The following are my concerns regarding the reproducibility statement of the paper.
1)	For all authors. . .
a)	Do the main claims made in the abstract and introduction accurately reflect the paper’s contributions and scope? [Yes]
i)	Confirmed.
b)	Did you describe the limitations of your work? [Yes]
i)	The limitations of the work are not clearly described. I highly recommend authors to add a new subsection and discuss the work limitations.
c)	Did you discuss any potential negative societal impacts of your work? [No] Does not apply.
i)	The paper does not include a broader impact statement. Please consider this section mandatory.
d)	Have you read the ethics review guidelines and ensured that your paper conforms to them? [Yes]
i)	The paper accessibility could be improved. To plot Figure 1 and Figure 2, please also use different shapes to differentiate lines.
2)	If you are including theoretical results...
a)	Did you state the full set of assumptions of all theoretical results? [N/A] Our works focuses on the experimental application of reinforcement learning for pruning strategy search.
i)	Confirmed.
b)	Did you include complete proofs of all theoretical results? [N/A] cf. above
i)	Confirmed.
3)	If you ran experiments. . .
a)	Did you include the code, data, and instructions needed to reproduce the main experimental results, including all requirements (e.g., requirements.txt with explicit version), an instructive README with installation, and execution commands (either in the supplemental material or as a URL)? [No]
i)	A sufficient material (code, dataset, and pre-train model) is provided to reproduce the results of the VGG model. However, I was unable to verify whether the code works since the code was only provided in the anonymous Github repository. Please upload the code in the source code on OpenReview.
b)	Did you include the raw results of running the given instructions on the given code and data? [Yes]
i)	Confirmed.
c)	Did you include scripts and commands that can be used to generate the figures and tables in your paper based on the raw results of the code, data, and instructions given? [Yes]
i)	The commands for generating the results of all tables and figures are not included in the code repository. Please consider you need to clearly mention all required commands separately for each table/figure of the paper in the README.md file.
d)	Did you ensure sufficient code quality such that your code can be safely executed and the code is properly documented? [Yes]
i)	The quality of the code is at an acceptable level. However, the quality can be significantly improved by (1) cleaning the code and removing unnecessary parts, (2) explaining the general functionality of each module with some comments, and (3) documenting all necessary instructions to regenerate the information reported in the tables and figures.
e)	Did you specify all the training details (e.g., data splits, pre-processing, search spaces, fixed hyperparameter settings, and how they were chosen)? [Yes]
i)	Please indicate in which section of the paper the experimental setup is described. In addition, I think some critical parameters have not been reported, such as the learning rate for training the models.
f)	Did you ensure that you compared different methods (including your own) exactly on the same benchmarks, including the same datasets, search space, code for training, and hyperparameters for that code? [Yes]
i)	Confirmed.
g)	Did you run ablation studies to assess the impact of different components of your approach? [No]
i)	Confirmed.
h)	Did you use the same evaluation protocol for the methods being compared? [Yes]
i)	Confirmed.
i)	Did you compare performance over time? [N/A]
i)	Confirmed.
j)	Did you perform multiple runs of your experiments and report random seeds? [Yes]
i)	In which part(s) of the paper are the results of multiple runs with different random seeds reported?
k)	Did you report error bars (e.g., with respect to the random seed after running experiments multiple times)? [Yes]
i)	Which part(s) of the paper does report error bars?
l)	Did you use tabular or surrogate benchmarks for in-depth evaluations? [No]
i)	Confirmed.
m)	Did you include the total amount of compute and the type of resources used (e.g., type of gpus, internal cluster, or cloud provider)? [Yes] Estimating the total amount of compute is difficult due to a multitude of different experiment not presented in the paper. However, we do specify the used hardware in the supplementary material.
i)	The hardware specification is not included in the supplementary material. Furthermore, I believe that the total compute time of the key experiments should be reported, since RL is slow in general, and readers should know the approximate overhead of optimization.
n)	Did you report how you tuned hyperparameters, and what time and resources this required (if they were not automatically tuned by your AutoML method, e.g., in a NAS approach; and also hyperparameters of your own method)? [N/A]
i)	The time and resources spent on automatic tuning hyperparameters are not reported.
4)	If you are using existing assets (e.g., code, data, models) or curating/releasing new assets…
i)	Confirmed.
5)	If you used crowdsourcing or conducted research with human subjects…
i)	Confirmed.


**Review Confidence:**

4: You are confident in your assessment, but not absolutely certain. It is unlikely, but not impossible, that you did not understand some parts of the submission or that you are unfamiliar with some pieces of related work.

**Review Rating:**

1: Strong reject

**Review Summary:**

N/A for reproducibility reviewers

---

### Meta-Review · Program_Chairs · 2022-05-11

**Recommendation:** Accept
**Confidence:** 5

**Metareview:**

After the discussion phase, the author's resolved (nearly) all problems regarding reproducibility (although the score in the review was not updated). In addition, one reviewer also increased the score to 5 in their reply. In view of the strengths and importance of the topic, I recommend acceptance.

---

### Decision · Program_Chairs · 2022-05-13

Accept